# Effect of Flue Gases’ Corrosive Components on the Degradation Process of Evaporator Tubes

**DOI:** 10.3390/ma14143860

**Published:** 2021-07-10

**Authors:** Mária Hagarová, Milan Vaško, Miroslav Pástor, Gabriela Baranová, Miloš Matvija

**Affiliations:** 1Institute of Materials and Quality Engineering, Faculty of Materials, Metallurgy and Recycling, Technical University of Košice, 04200 Košice, Slovakia; maria.hagarova@tuke.sk (M.H.); gabriela.baranova@tuke.sk (G.B.); milos.matvija@tuke.sk (M.M.); 2Department of Applied Mechanics, Faculty of Mechanical Engineering, University of Žilina, 01026 Žilina, Slovakia; 3Department of Applied Mechanics and Mechanical Engineering, Faculty of Mechanical Engineering, Technical University of Košice, 04200 Košice, Slovakia; miroslav.pastor@tuke.sk

**Keywords:** municipal waste incinerator, corrosion, steel surface, flue gas, chlorides, feedwater

## Abstract

Corrosion of boiler tubes remains an operational and economic limitation in municipal waste power plants. The understanding of the nature, mechanism, and related factors can help reduce the degradation process caused by corrosion. The chlorine content in the fuel has a significant effect on the production of gaseous components (e.g., HCl) and condensed phases on the chloride base. This study aimed to analyze the effects of flue gases on the outer surface and saturated steam on the inner surface of the evaporator tube. The influence of gaseous chlorides and sulfates or their deposits on the course and intensity of corrosion was observed. The salt melts reacted with the steel surface facing the flue gas flow and increased the thickness of the oxide layer up to a maximum of 30 mm. On the surface not facing the flue gas flow, they disrupted the corrosive layer, reduced its adhesion, and exposed the metal surface. Beneath the massive deposits, a local overheating of the inner surface of the evaporator tubes occurred, which resulted in the release of the protective magnetite layer from the surface. Ash deposits reduce the boiler’s thermal efficiency because they act as a thermal resistor for heat transfer between the flue gases and the working medium in the pipes. The effect of insufficient feedwater treatment was evinced in the presence of mineral salts in the corrosion layer on the inner surface of the tube.

## 1. Introduction

In the heating industry, a boiler device is a whole group of devices used to convert the chemical energy contained in a solid fuel such as coal, biomass, or municipal waste, into thermal energy in the form of water or steam. It is further used to drive steam turbines or in the form of hot water supply [1,2]. The specific conditions of corrosion in thermal power devices result from the fact that while the outer surface of heat-exchange tubes is affected by the oxidizing effect of the flue gases or other combustion products with different temperature levels, the inner surface is affected by the oxidizing effect of feedwater, saturated or superheated steam. High-temperature corrosion and low-temperature corrosion are priority degradation processes in the operation of a boiler device [3,4]. The boiling system of the boiler represents an evaporator in which the phase conversion of the feedwater into saturated steam takes place by heat transfer from the flue gases through the heat-exchange surface areas of the water wall tubes. The material of the water wall tubes differs depending on the temperature of their outer surface. In the case of a lower temperature load, steel W. Nr. 1.1148 or steel W. Nr. 1.0345 is sufficient. For higher temperature loads, steels W. Nr. 1.7380, W. Nr. 1.7715, or steel W. Nr. 1.7383 are used, for example. The groups of steels noted above differ in their strength properties at higher temperatures. Their resistance to oxidation in the flue gas environment is comparable [5,6]. For demanding operating conditions where high corrosion resistance is required while maintaining strength in the creep area, heat-resistant steels P91, P92, or martensitic 9Cr MarBN steels are used [7]. The failure-free operation conditions of the evaporator are related to: the properties of the steel material of the evaporator tubes (strength at elevated temperature); the temperature and the composition of flue gases acting on the outer surface of tubes; heat transfer conditions through tube walls; and last but not least the properties of feed (boiler) water, which affects the condition of the inner surface.

The high corrosion resistance of steel is based on the formation of a sufficient protective layer of Fe oxides: Fe_3_O_4_ magnetite and Fe_2_O_3_ hematite, both on the outside and inside the tubes. The municipal waste incineration results in a high concentration of corrosive and deposit-forming elements. Under real operating conditions, compact oxide layers are not formed; there are cracks, bulges, and other defects [8]. The presence of nodules can lead to rapid growth in the form of an oxide layer without the protective effects of the steel base. In the literature, this effect is well known as “breakaway” oxidation [9].

Temperature changes, mechanical deformation at the oxide/substrate interface, and changes in structure due to the long-term temperature effect promote the initiation of thermal fatigue, which ultimately shortens the service life of materials. Replacement of conventional structural steels with high-strength steels will cause a significant change in the cross-section of individual members without changing their load-bearing capacity [10,11]. Generally, at the beginning of the oxidation process, a certain critical amount of an element (e.g., Cr) must be present in the steel to form a protective oxide layer. If this level is below the critical value, the layer protective effect is lost in the cracks, and there is a rapid loss of steel material. The high degree of metallic material degradation will prevent its long-term use [12,13]. The main corrosive element on the inner surface of heat-exchange tubes is oxygen. Its solubility depends on the boiler feedwater temperature. When assessing the risk of corrosion, it is necessary to know the amount of oxygen that enters the boiler water system. In the range of feedwater temperature of 85 to 95 °C, the oxygen content is reduced to about 2.3 to 1.0 ppm, according to [14]. On the other hand, if the feedwater temperature is too high, cavitation may occur, for example, on the impellers of the feed pump. Inhibitors are used to reduce the oxygen content in operating conditions. Sufficient treatment of the feedwater ensures the formation of a protective layer that sufficiently slows down the corrosion process on the steel surface. If the dosage (or concentration) of inhibitors is insufficient, it can lead to the formation of oxygen corrosion by the mechanism of deep corrosion pits formation. Corrosive contaminants such as oxygen, chloride, sodium, sulfate, copper, and silica found in excessive concentrations induce considerably high conductivity. Regular monitoring of their levels is greatly important to prevent corrosion and reduce the formation of deposits on heat-exchange surfaces. The introduction of the optimal number of diagnostic parameters, as well as establishing adequate frequencies between measurements and utilizing corresponding analytical tools, lead to improved chemical monitoring of feedwater [15,16]. Corrosion problems are usually associated with operational problems and equipment maintenance, leading to recurrent partial problems, respectively shutdown [17]. The water treatment process begins with an initial purification and filtration step in which suspended solids are separated from the water. In the next step, by heating the feedwater, the solubility of the dissolved gas is reduced so that the water is de-aerated. The residual oxygen is removed by adding oxygen scavengers. The most used oxygen scavengers for boilers are sodium sulfate and hydrazine. Phosphates are used to control the deposit formation [18,19]. The presence of dissolved oxygen is considered a key cause of feedwater corrosivity. It is, therefore, necessary to minimize this risk to an acceptable level. By optimal treatment of the feedwater, we achieve the formation of a passive magnetite layer on the inner metal surface, which provides increased protection against corrosion [20]. The quality of the corrosion layer on the outer surface of steel tubes is influenced mainly by the composition of corrosion products arising in the combustion chamber of the boiler, which depends on the type of fuel used. The corrosion rate in the boilers for the combustion of municipal waste is higher compared to conventional coal-fired power plants operating at higher temperatures. High rates of corrosion are related to the heterogeneous nature of the fuel and its variable chlorine content. The heterogeneous nature of the fuel makes it difficult for operators to maintain uniform combustion conditions that are provided (more complied with) in steam boilers. Municipal waste contains alkali metals such as sodium and potassium, heavy metals such as lead, zinc, and various chlorine-containing compounds, all of which can form potentially corrosive agents [21,22,23]. The active compounds noted above are part of the deposits and, at the same time, act as corrosion stimulators (mainly chlorides). The composition of the fuel, along with the operating parameters, influences the fuel gas composition as well as the composition and the properties of the deposits on the surfaces of the heat-exchange tubes. Other factors such as high surface temperature of the evaporator and flue gas also affect the high-temperature corrosion. The temperature gradient between the flue gas temperature and the metal surface determines the vapor condensation, the settling rate, and the composition of the corrosion layer. The presence of lead and zinc in the layer reduces its melting point and contributes to enhancing the effect of chlorides in high-temperature corrosion in boilers [21].

Chlorine (e.g., as part of HCl (g)) can penetrate through a protective oxide layer on the steel surface and participate in a corrosion reaction to form FeCl_2_. The formed chloride sublimes and decomposes on contact with oxygen:(1)$$3FeCl_{2}\left(g\right)+2O_{2}\to Fe_{3}O_{4}+Cl_{2}$$
possibly on
(2)$$2FeCl_{2}\left(g\right)+\frac{3}{2}O_{2}\to Fe_{2}O_{3}+Cl_{2}$$

The cycle can be repeated to maintain the oxidation of the metal surface under the non-protective oxide layer [22]. The other influential component of flue gases is SO_2_. Sulfur oxide reacts with alkali metal chlorides to form sulfates. The formed sulfates are less corrosive. Sulfur compounds are often added during the combustion process to turn alkali metal chlorides into less corrosive alkali sulfates. The authors [24] found a synergistic effect on the corrosion process in the case of Cl content in the corrosion layer and SO_2_ in the flue gas environment. Low-melting eutectics NaCl-FeCl_2_, respectively KCl-FeCl_2_—whose melting point (eutectic temperature) is 474 °C, respectively 451 °C—in the existing liquid form, prevent the formation of a compact protective layer of corrosion products [25]. Any causation leading to a loss of stability of the magnetite layer and causes failures of its integrity will later evince itself in a gradual loss of integrity of the tube wall. In this case, thick layers of Fe oxides are formed on the surfaces of the tube, there is extreme overheating of the walls, and an increase in the oxidation rate in the maximum stress planes, which lead to the final failure of the tubes by perforation. Even long-term overheating of the boiler heat-exchange tubes in the flue gas environment with the calculating temperature (the allowable tube wall temperature) of the tube wall increases the oxidation rate and at the same time worsens the mechanical properties. At the same time, the allowable stress value decreases, and the physical properties deteriorate [26]. As a result, the tubes are unable to withstand the internal pressure of saturated steam, which in the evaporators is at the level of ~25 MPa, and cracking occurs [27]. In addition to the above effects on the service life of the steam boiler tubing system, the monitoring of operating parameters also plays an important role, as in the cause of failure of protective mechanisms, unexpected temperature processes may occur, which can subsequently cause a device accident [28].

Although extensive research on corrosion in municipal waste incineration plants (WIP) is published, it mostly concerns the assessment of the impact of high-temperature corrosion on the degradation mechanism of superheaters. In this study, it proved necessary to study the chemical composition of deposits on the evaporator due to the severity of high-temperature and low-temperature corrosion, which led to the heat-exchange tube perforation and shutdown of the evaporator part of the device. This information should prove useful in taking anti-corrosion arrangements by optimizing the proportion of aggressive components (HCl) in the fuel and subsequently in the combustion environment. It would lead to a reduction in scale formation and corrosion in the WIP.

## 2. Materials and Methods

The evaporator is a system of tubes in which the heat from the flue gas is used to heat mediums (saturated steam and water). As a consequence of the high temperature, intense oxidation and loss of wall material of evaporator tubes often occurs. If the temperature field is situated on the side of the medium on the outer and inner wall of the tube, the chemical corrosion process intensifies to the bounded area. Figure 1 shows a schematic layout of a boiler device with an indication of the location of the evaporator.

The operating parameters of the boiler were as follows: the boiler nominal capacity of 30 t/h, superheated steam pressure 1.98 MPa, the steam temperature in the evaporator tubes 270 °C. The analyzed evaporator tube was made of steel W. Nr.1.1148 after normalizing heat treatment. This steel is used to produce pressure parts of energy devices. The dimensions of the seamless tube were 51 × 5 mm. The flue gas temperature at the inlet to the evaporator was reaching 650 °C (Figure 1). Standard values of strength characteristics are given in Table 1.

An inspection was carried out on the device serving for more than 30 years and revealed perforations on the evaporator tube.

A tensile test was used for the determination of the mechanical properties of steel. Microscopic analysis, macroscopic analysis, and analysis of the chemical composition of the material and corrosion layer were performed to determine the causes of formation of the steel tube perforation. First, a tensile test was performed, which confirmed the use of the material declared by the manufacturer. The determination of mechanical characteristics, namely, yield strength Rp0.2 and steel tensile strength Rm, was performed according to EN ISO 6892-1: 2009 [29,30] using a testing machine with a load of 0 ÷ 200 kN and a speed of 1 mm min^−1^ (Table 2). The mechanical properties of the material from the selected tube were realized on 3 test samples.

The starting point of the analyses was macroscopic analysis on a stereomicroscope. In the next step, microscopic metallographic analysis was used to determine the condition of the structure of the taken material and the nature of its degradation. The analysis was performed on a light microscope. The samples for microscopic study were prepared by standard metallographic procedures (grinding, polishing, and etching).

The chemical composition of the tube material and the corrosion layer on its outer and inner surface was determined by energy dispersive (EDX) analysis using an EDX analyzer on a scanning electron microscope JEOL JSM 7000F. Part of the obtained data was also processed and analyzed using algorithms programmed in the MATLAB software package.

## 3. Results

### 3.1. Macroscopic Analysis and Dimensions of the Damaged Tube

For macroscopic analysis, the small parts were cut from the evaporator tube, with different types of damage nature, and marked as 1, 2, 3, and 4 (Figure 2). Marking of the samples with the description of analyses is presented in Table 3.

During the combustion process, a thick deposit layer was formed, located on the outer surface of the evaporator tube (Figure 3). This deposit layer was embossed, non-uniform, with different colorations (e.g., light-ocherous coloration corresponds to ash deposit).

The inner surface of the tube from the cutout in Figure 3a (below the deposit layer) is shown in Figure 4. The color of the corrosive layer formed on the inner surface is rusty brown in the central part, containing the highest proportion of Fe_2_O_3_, and at the edges is formed by a magnetite layer of Fe_3_O_4_. While in the lower part of the tube (on the left in Figure 4a), the corrosion layer was well adhered, in the upper part of the tube, the corrosion layer was separated from the surface (on the right in Figure 4b). Under the deposit, the steel material was overheated due to the deterioration of heat transfer through the tube wall, resulting in a loss of magnetite adhesion to the inner surface of the water wall tube.

In Figure 5 is a documented inner surface located below the deposit on cutout 2 (Figure 3b). A significant proportion of hematite is visible on the lower part of the tube, while on the upper part, there is mainly magnetite with a surface formed by mineral deposits with a white-gray coloration based on Ca, respectively Mg. This fact was also confirmed by EDX analysis. The figure shows the interface created by the condensation of water vapor during the shutdown.

Cutouts 3 and 4 were prepared from areas with the large extent perforations (Figure 6a) and the small extent perforations (Figure 6b) of the tube wall. The small perforation was covered with a well-adhered corrosion layer on the steel surface, as can be seen in Figure 6a. Therefore, its presence was not detected even with such a size. The large perforation was detected not only because of its extended size but also due to the loss of adhesion of the corrosion layer on the surface.

Figure 7 shows the inner surface of the cutout 3 of the tube from a place with a perforation of a small extent (Figure 6a). The image shows a dark gray, very embossed magnetite corrosion layer with pitting defects that can extend to the steel base. This layer is covered with a rusty brown colored hematite in several areas. Areas with light gray to white coloration (mineral salt content) indicate insufficient water treatment (Figure 7b).

The tube damage of a large extent (Figure 6b) occurred on its upper part, at the place of the smallest wall thickness of the tube (Figure 2, place 4). The surface of the tube was in this area without the cover of corrosion products of greater thickness. Figure 8 documents an ununiformed tube wall thickness with a minimum measured wall thickness of 0.9 mm (the original tube wall thickness at the beginning of operation was 5 mm).

Detailed images of the lower part of the tube, the cutout 4 (according to Figure 2), from the place of a large extent tube wall perforation are shown in Figure 9. The inner surface is covered with a magnetite layer, the thickness of which is the largest on the bottom surface of the tube. Areas with a light gray to white color (mineral salt content) indicate insufficient feedwater treatment. The area with the different morphology of the corrosion layer represents the interface of the condensed water vapor level (Figure 9b).

### 3.2. Microscopic Analysis

Three samples, marked T1, T2, and T3, were taken from cutout 4, i.e., from a place with a large wall perforation for analyzing the structure of the evaporator tube and the nature of its damage. Sampling points, as well as the indication of analyzed areas, are clear from Figure 10.

In Figure 11 is documented the structure of steel with ferrite (light grains) and pearlite (dark grains) analyzed on sample T1. The steel structure is relatively uniform in terms of grain size and distribution of structural components and corresponds to the declared steel W. Nr. 1.1148 after normalization heat treatment. The structure refinement was visible in the vicinity of the crack, in the part of the tube where the highest decrease in the wall thickness was found, down to a thickness of 0.9 mm. This structure refinement is the result of an abrasive wear process, which caused the loss of material from the upper part of the outer tube surface, and at the same time during this process, there was a strain hardening of the structure. The structure refinement was observed along the entire crack edge and in the structure bounding the crack, as is documented in Figure 11.

Figure 12 shows the place of the beginning of the crack extension on the sample T2 (in Figure 10, this location is indicated with a white outline). The surface of the crack was covered with a corrosive layer, especially during its extension in the material. A slightly deformed structure, which is color-marked in the figure, was observed in the vicinity of the corrosion layer. Figure 13 shows sample T3 with the edge of the crack separating parts of the material. Gradual polishing reduced the distance of the separated parts of the material due to the wedge-shaped crack growth towards the inside of the steel tube wall.

In Figure 14 is documented the sample T3′s outer and inner surfaces with the corrosion layer, in the cross-section of the evaporator tube, outside the area of the propagating. After abrasive wear from the flue gas side, only a part of the corrosion layer remained attached to the outer surface. Its good adhesion was also ensured by the proportion of contained mineral components (which was also confirmed by EDX analysis), which with the Fe oxides formed a hard “shell” on the surface. The inner surface is covered with a relatively thick corrosion layer. Its growth has intensified because of the material’s local overheating as its thickness decreases.

### 3.3. EDX Analysis of the Corrosion Layer

Figure 2 contains samples taken for scanning electron microscope (SEM) analysis. They were taken from cutouts 1, 2, and 3 (according to Figure 2). Brief description of the analyzed samples:Cutout 1—the place with a layer of massive deposits (marking of samples: 1SEM—the upper part of the tube and 2SEM—the lower part of the tube);Cutout 2—the place with a layer of deposits on the outer surface (marking of sample: 3SEM—cut from the top of the tube);Cutout 3—marking of sample 4SEM—obtained by separating the deposit layer from the lower part of the tube.

SEM analysis was performed on each outer and inner surface of the tube samples. Figure 15 documents the outer surface of the 1SEM sample (according to Figure 2), and Figure 16 and Figure 17 show the qualitative SEM analysis in the form of Spectrum 4 and 5. Summary quantitative analysis of the chemical composition of all analyzed outer and inner surfaces of the samples according to Figure 2, is shown in Table 4.

## 4. Discussion

Corrosion damage to the evaporator tube was not uniform around the circumference of the tubes, nor concerning the distance from the sampling point. It can be stated that the most significant corrosion occurred on that part of the outer circumference which was facing the flue gas flow and was exposed to higher temperatures. According to [31], oxides, chlorides, silicates, and others are present in the corrosion layer in addition to sulfates and chlorides. The composition of the deposits changes with the location of the heat-exchange tube in the device due to the influence of the device arrangement geometry (in our case, the evaporator), the flue gas flow rate, and the temperature on the metal surface.

According to [32], the composition of products and residues is influenced by the composition of the input waste. The flue gases from waste incineration contain gaseous HCl, SO_2_, as well as solid particles, which at higher temperatures form ash deposits on the boiler pipes. These deposits contain chlorides, oxides, and sulfates of Na, K, Pb, and Zn metals. The intensity of corrosion in WIP depends on many factors, in particular the presence of chlorine-containing compounds and the formation of deposits. In general, the corrosion rate increases with the concentration of Cl in the input fuel, flue gas, and ash up to a certain concentration level above which there is no further increase of corrosion. On the outer surface of the analyzed evaporator tube at the sampling point at the distance of approximately 400 mm from the bend on the upper part of the tube, there was a corrosion layer (a deposit layer) with heterogeneous composition of individual layers after sediment cross-section. The layer of light-ocherous color was dominant, which partly covered the dark gray layer, or a rusty brown layer of Fe oxides. The maximum layer thickness reached approximately 30 mm. At a greater distance from the edge, a well-adhered corrosion layer on the surface had a larger share, the thickness of which reached a maximum of 8 mm. At a length of approximately 600 mm from the bend, the surface of the tube was covered with fractions of the corrosion layer. The Fe and O content was found in the corrosion layer on the outer surface of the evaporator tube as a part of the corrosion layer.

Chlorine, sodium, and potassium contributed to the formation of chloride compounds. The highest content of the mentioned elements was found on the outer surface, both on the upper and lower part of the tube on samples 1SEM and 2SEM (according to Figure 2 and Table 4), with the maximum Cl value approximately 30.0 wt % and Na approximately 15.0 wt %. The potassium content was the highest on the upper part of the tube on the 1SEM sample on its outer surface and its values were at a maximum level of approximately 14.5 wt %. Sulfur can pass through the imperfections of the protective oxide layer to the metal surface and form a sulfide layer, which is not only porous but can eventually cause the entire protective layer of Fe oxides to peel off. The highest sulfur content was found on the outer surface at the bottom of the tube in the 2SEM sample, specifically 11.9 wt %. In other parts of the analyzed outer surface of the tube, its content was up to approximately 3.9 wt %. The layer also included heavy metals Zn and Pb, which lower the melting point of the corrosive layer on the metal surface. Chlorides of these metals have a low melting point and can cause melting of deposits even at temperatures of up to 300 °C [22,25]. EDX analysis showed a relatively high content of Zn and Pb in all analyzed samples, with the most on sample 1SEM with a maximum value of about 4.0 wt % and 4SEM, the sample from the bottom of the tube, where the surface has the morphology of the flowing melt after solidification (Figure 2).

The study [33] presents the concentrations of some corrosive components in the flue gases of boilers for municipal waste incineration. The HCl concentration was in the range of 560–2240 ppm, and the SO_2_ concentration 100–2000 ppm. Deposits from three different municipal waste incinerators showed significant amounts of heavy metals, namely, Pb^+^ from 1.6 to 7.5 wt %, Zn^2+^ from 2.3 to 9.7 wt %, and chlorides in the form of Cl^−^ in the range 0.1–1.2 wt %.

According to [34], all chlorine compounds in municipal solid waste (MSW) are converted to hydrogen chloride in the combustion environment. The amount of HCl in the flue gas from the waste incinerator ranges from about 400 to 1500 ppm, based on the general rule that each 0.1% Cl in the waste generates about 80 ppm HCl.

As the flue gas flows out of the combustion chamber (in the combustion chamber space), the flue gas temperature decreases, and the volatile alkali chlorides condense. Mixtures of salts based on chlorides (eutectics) NaCl-FeCl_2_, respectively KCl-FeCl_2_ have a melting point (eutectic temperature) of 374 °C, respectively 393 °C, which can cause the deposits to melt at a flue gas temperature of approximately 650 °C.

As reported by the authors in [35], the deposit on the surface of the heat-exchange tubes contained large amounts of Ca and S, indicating the presence of CaSO_4_. In the case of the high content of Na, K, Cl, and S, it is the occurrence of a eutectic of alkali chloride, respectively sulfide. A large amount of sulfur in deposits was due to the sulfation of alkali chlorides at high steam temperature forming thin and dense alkali sulfates layer at the metal/oxide interface.

The eutectics mentioned above in the existed liquid form prevent the formation of a compact protective layer of corrosion products and at the same time create a precondition for the course of electrochemical corrosion because the liquid phase is an electrolyte [24,36].

Alkaline compounds in the form of fly ash abrasively damage the tube surface during the flue gases’ flow in the working space, which leads to a decrease in tube wall thickness to values where the steel strength drops below the calculated strength value, and the material degrades with a cracking formation. In the areas where the surface of the analyzed evaporator tube was affected by abrasive wear, there were two perforations in the thinnest part of the tube wall: the first of small extent and the second of large extent (Figure 2). Microscopic analysis of the steel showed that in the vicinity of the perforation (crack) in Figure 11 and Figure 12, the structure was finer compared to the structure where the wall of the tube was not thinned. This change in the structure took place in the process of abrasive wear by solid flue gas particles, which caused both the loss of material and its strain hardening [37]. Due to the uneven flue gas flow in the space between the water wall tubes, their surface was covered with a non-uniform deposit thickness. In the place where corrosion deposits were separated due to loss of adhesion, together with the flue gas became abrasive at another place on the tube surface (cutouts 3 and 4 in Figure 2).

Analysis of the inner surface showed a difference in the corrosion effect of the surface. At the bottom of the tube was a layer of dark gray magnetite lined with rusty brown hematite at the edges. The magnetite included particles that could degrade the layer quality (Figure 9). On the inner surface of the tube, a visible interface was formed during the condensation of water vapor during downtime (Figure 4, Figure 5, and Figure 7). The content of mineral salts in the places with light gray to white coloration indicated insufficient treatment of the feedwater.

In the case of EDX analysis of the inner surface of the evaporator tubes, the presence of a corrosive layer of Fe oxides was detected. In the corrosion layer (except for Fe and O), Ca, Na, Si, and Mg, which were part of the mineral deposits, were also analyzed. According to EN 12952-12, the feedwater of boilers should contain up to 0.02 mmol L^−1^ Ca + Mg and Na ^+^ + K ^+^ up to 0.010 mg L^−1^ at a feedwater injection pressure up to 2.0 MPa.

The study [38] has demonstrated that corrosion of boiler tubes in waste incinerators can be interpreted as fused salt corrosion, and the corrosion rate of steels and alloys is influenced primarily by the volume fraction of fused salts at the corrosion temperature. When the volume fraction of fused salts is high, corrosion is expected to be severe. Concentration of zinc and sodium compounds in the deposits results in enhancing fusion of salt mixtures, and hence, can accelerate corrosion of carbon steel. The corrosion rate of boiler tubes may be predicted, to a certain extent, through knowing the deposit chemistry in real waste incinerators.

## 5. Conclusions

Based on the performed analysis, the effect of the chlorine, as a corrosive component, present in the fuel on the degradation of the evaporator steel tube outer surface was detected. In the combustion process, the superposition of Cl and its compounds manifested itself in the reaction of flue gases with the steel surface by the following mechanisms:The formation of a heterogeneous corrosion layer was supported by a high chlorine content of 30.0 wt %, which negatively affected its compatibility and reduced its protective effect. The massive corrosive layer was also disturbed by a high content of approximately 15.0 wt % Na and approximately 14.50 wt % K, assuming the formation of alkali metal chloride supporting the degradation process.The formation of low-melting eutectics with a high heavy metals proportion, with almost 6.0 wt % Zn and 3.5 wt % Pb, which flowed by gravitational forces into the lower part of the tube and disturbed the protective corrosion layer on the tube’s outer surface.The effect of solid products forming in the combustion process, which became part of the flue gas, and the abrasive wear occurred on the outer surface resulting in the perforation of the wall of the tube of large and small extents.

In the combustion process, the chlorine present in the fuel becomes an active corrosive component in various forms and thereby increases its effect several times on the surface degradation of the heat-exchange tubes of WIP.

To ensure failure-free operation of the evaporator, it is necessary to optimize the chemical composition of the combustion environment by separating the incinerated waste, mainly by limiting the amount of recyclable plastic packaging with the effect of reducing HCl and heavy metals in flue gases.

## Figures and Tables

**Figure 1 materials-14-03860-f001:**
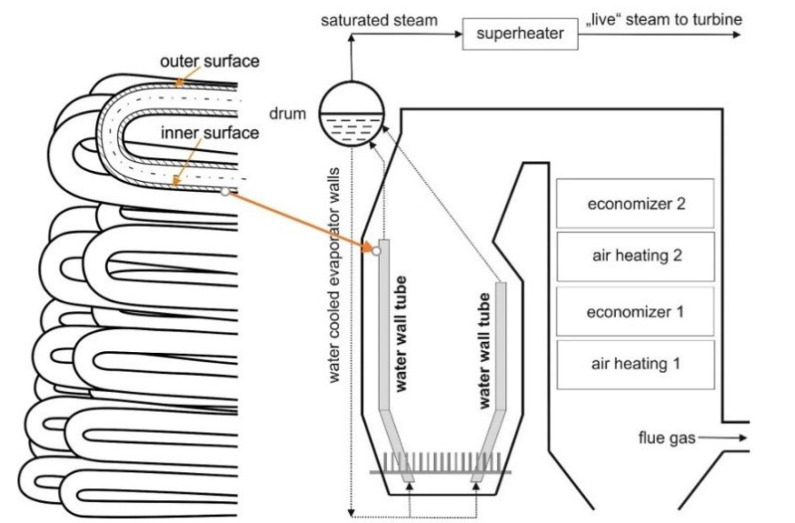
Scheme of the water wall tubes (the evaporator tubes) location in the combustion chamber of the boiler.

**Figure 2 materials-14-03860-f002:**
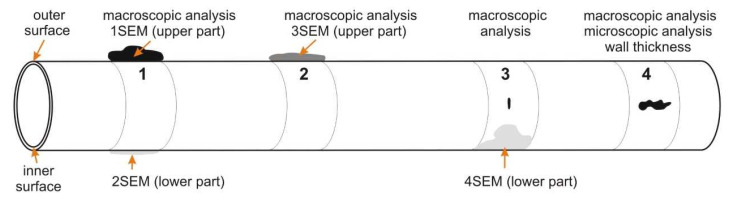
Schema of evaporator tube with the location of cutouts assigned for analyses according to Table 3.

**Figure 3 materials-14-03860-f003:**
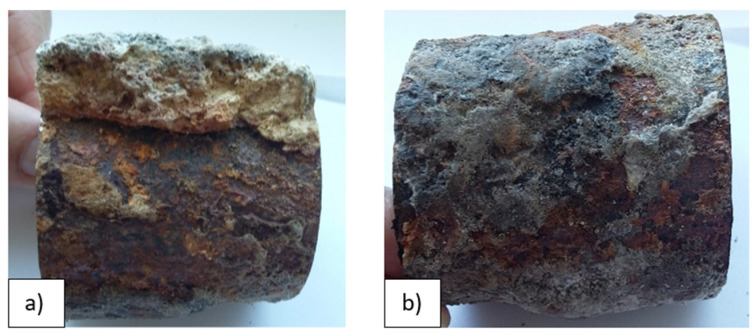
The cutout from the evaporator tube: (**a**) massive deposit layer on the cutout 1; (**b**) deposit layer on the cutout 2 (according to Figure 2).

**Figure 4 materials-14-03860-f004:**
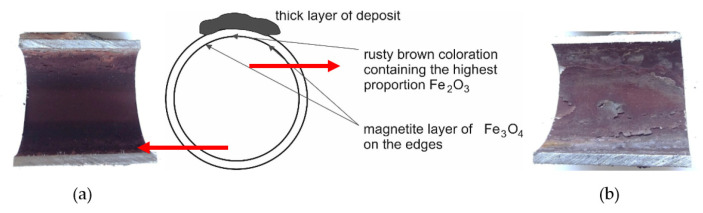
The inner surface of the cutout 1 tube with a massive deposit: (**a**) the lower part of the tube; (**b**) the upper part of the tube placed under the deposit layer according to Figure 3a.

**Figure 5 materials-14-03860-f005:**
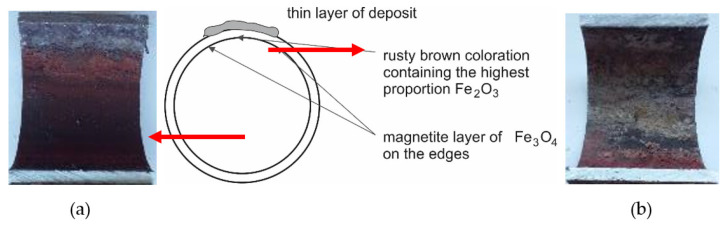
The inner surface of the cutout 2 tube with the deposit: (**a**) the lower part of the tube; (**b**) the upper part of the tube placed under the deposit according to Figure 5b.

**Figure 6 materials-14-03860-f006:**
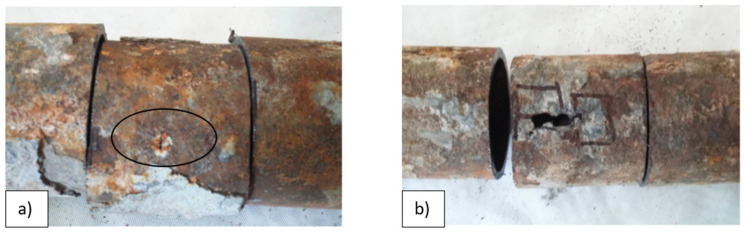
(**a**) Cutout 3 with a place of perforation of a small extent; (**b**) cutout 4 place of perforation of a large extent.

**Figure 7 materials-14-03860-f007:**
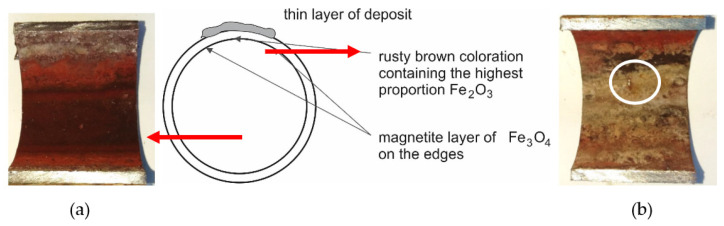
The inner surface of cutout 3 with the deposit: (**a**) the lower part of the tube; (**b**) the upper part of the tube placed under the deposited layer according to Figure 6a.

**Figure 8 materials-14-03860-f008:**
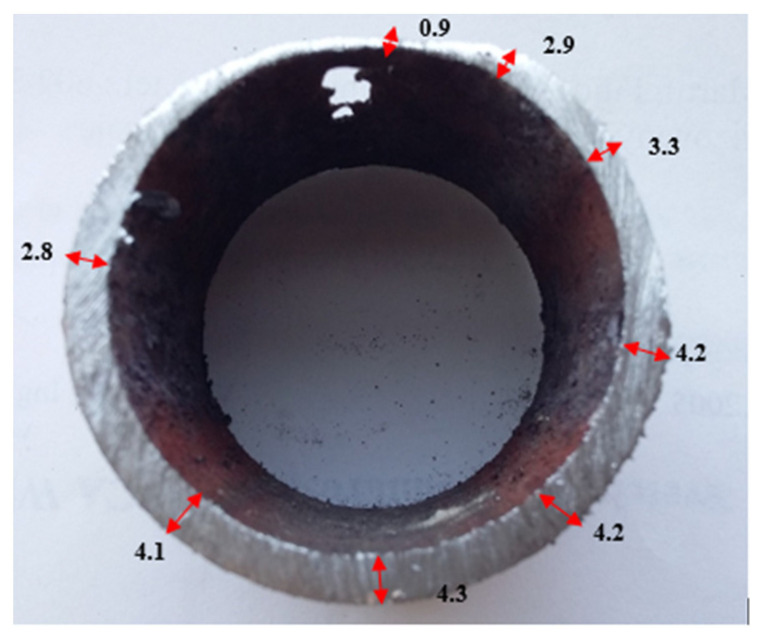
The wall thickness of the evaporator tube at the perforation area; wall thickness (in mm).

**Figure 9 materials-14-03860-f009:**
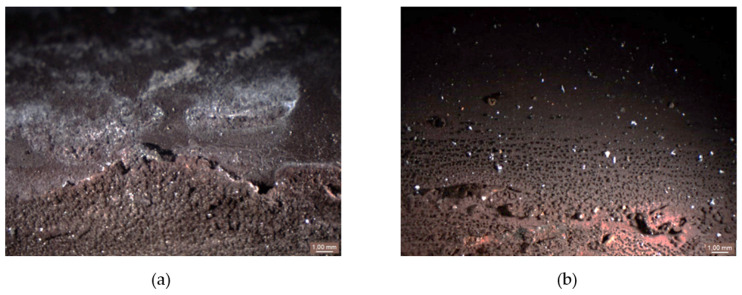
(**a**) The overall view of the inner surface of the tube bottom part from Figure 8. This is cutout 4 with a perforation of a large extent, according to Figure 2; (**b**) the corrosion layer on the interface of the condensed water vapor level.

**Figure 10 materials-14-03860-f010:**
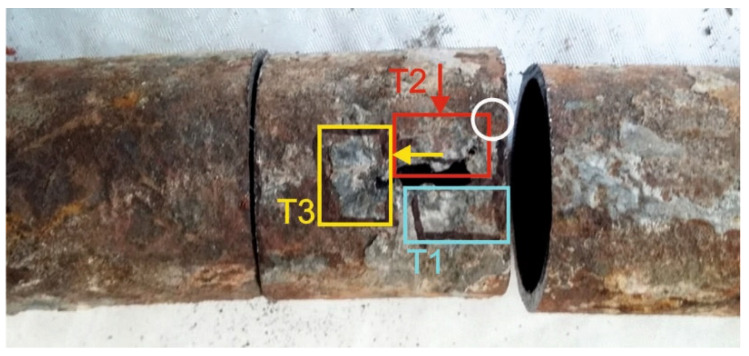
Sampling points T2, T3 with observed areas (indicated by arrows). On sample T1, the observed area was parallel with the outer surface of the tube.

**Figure 11 materials-14-03860-f011:**
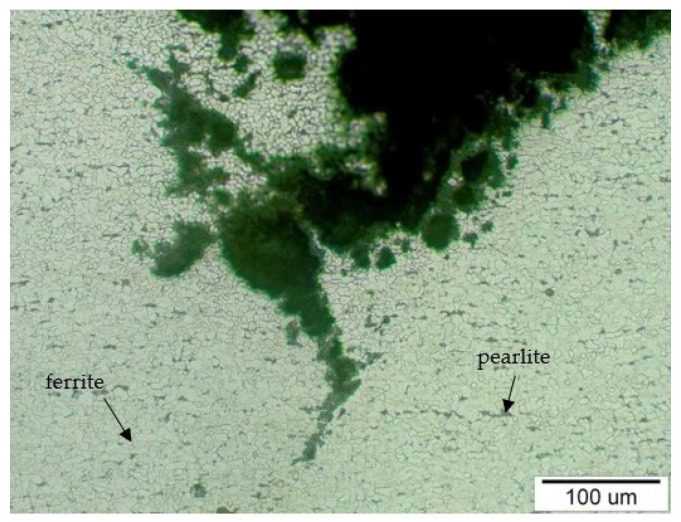
The steel structure corresponded to steel W. Nr. 1.1148 (consisting of the light ferrite grains and the dark pearlite grains) with a crack crossing into the material, sample T1 (etch, Nital).

**Figure 12 materials-14-03860-f012:**
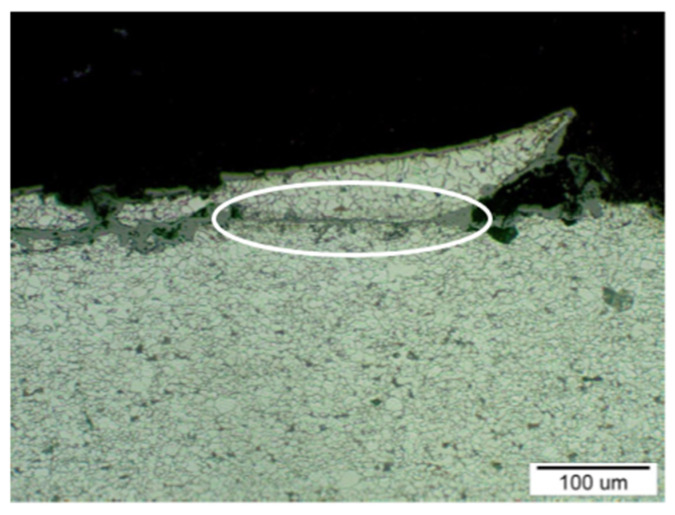
The area with the beginning of the crack extension in the steel structure, according to Figure 10, where it is marked with a white outline; etch, Nital.

**Figure 13 materials-14-03860-f013:**
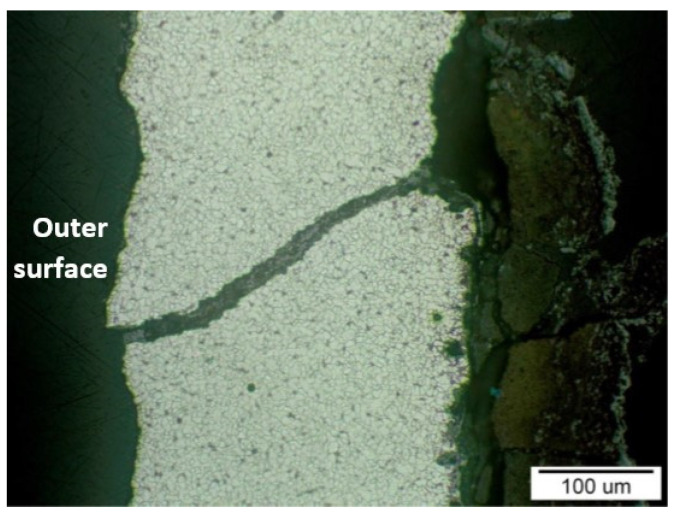
The development of the crack towards the inside of the material with the corrosion layer on the inner surface of the sample T3 (in Figure 10 is the observed area indicated by the yellow arrow); etch, Nital.

**Figure 14 materials-14-03860-f014:**
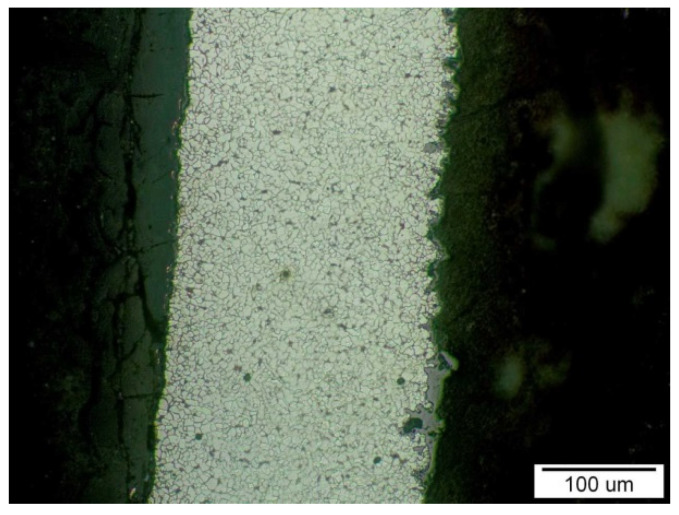
The outer surface (left) and the inner surface (right) of the T3 sample with a corrosion layer on the inner surface. On the outer surface is a visible part of the corrosion layer that remained attached to the surface; etch, Nital.

**Figure 15 materials-14-03860-f015:**
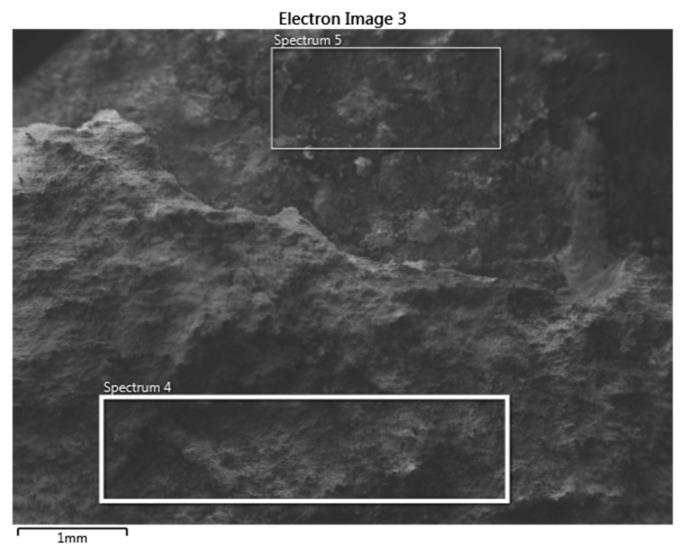
The outer surface of 1SEM sample with analyzed Spectrum 4 and 5 areas; SEM.

**Figure 16 materials-14-03860-f016:**
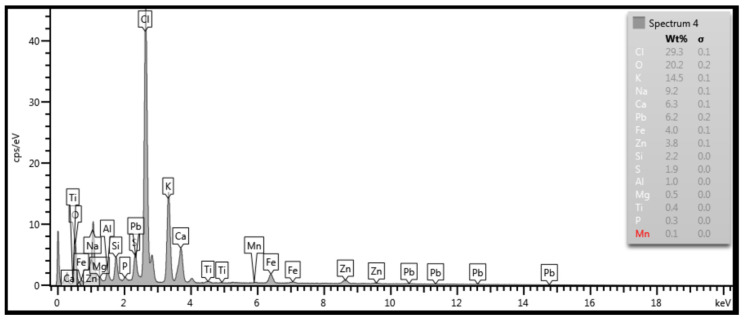
Spectrum 4 according to Figure 15; SEM.

**Figure 17 materials-14-03860-f017:**
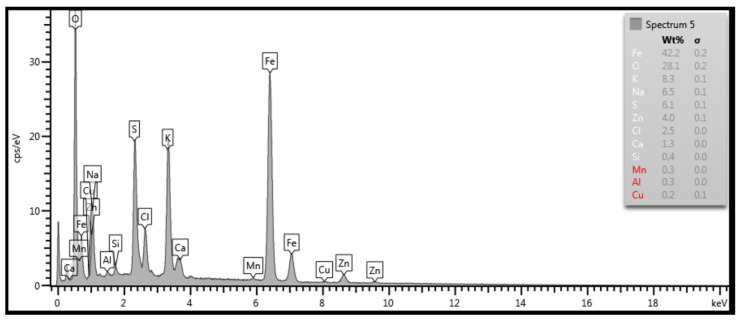
Spectrum 5 according to Figure 15; SEM.

**Table 1 materials-14-03860-t001:** Standard strength characteristics of steel W. Nr. 1.1148.

W. Nr. 1.1148 Steel	Strength Characteristics According to the Standard W. Nr. 1.1148					
Characteristics	Rp0.2					Rm (MPa)
Temperature (°C)	20	300	350	400	450	340–470
Min. value (MPa)	235	147	127	107	88	

**Table 2 materials-14-03860-t002:** Real mechanical parameters of W. Nr. 1.1148.

W. Nr. 1.1148 Steel	Mechanical Parameters of Steel Determined by Tensile Test According to EN ISO 6892-1: 2009	
Parameter	Rp0.2	Rm
Value (MPa)	282	401

**Table 3 materials-14-03860-t003:** Marking of evaporator tube cutouts for analysis and corrosion test.

Cutout Number	Localized Place	Analysis/Test
1	massive deposit on the outer surface of the tube	macroscopic, microscopic, and EDX analysis, corrosion test
2	layer under the deposit	macroscopic and EDX analysis
3	perforation of a small extent	macroscopic analysis
4	perforation of a large extent	macroscopic and microscopic analysis, wall thickness measurement

**Table 4 materials-14-03860-t004:** Quantitative EDX analysis in wt %.

Element	Spectrum										
	1SEM Outer			2SEM Outer		3SEM Outer	4SEM Outer	1SEM Inner	2SEM Inner	3SEM Inner	4SEM Inner
	1	4	5	6	7	8	12	9	10	11	13
O	19.46	20.24	28.06	41.55	31.55	37.50	47.31	30.73	31.9	35.08	43.91
Na	15.12	9.20	6.47	3.61	12.50	2.31	1.36	0.91	1.22	0.77	2.65
Al	1.04	1.03	0.26	2.36	1.86	2.17	1.69	0.25	0.30	0.17	3.05
Si	1.93	2.17	0.42	3.07	2.75	4.06	2.11	3.24	2.89	5.79	17.58
P	0.25	0.32	-	0.58	1.22	1.51	0.57	3.87	3.84	3.88	1.22
S	1.35	1.91	6.07	11.89	6.26	3.85	0.97	0.11	0.41	0.12	1.46
Cl	30.48	29.27	2.48	3.42	12.56	2.21	3.96	-	0.32	0.09	3.66
K	12.68	14.46	8.25	1.40	1.12	1.33	0.72	0.14	0.31	9.17	0.78
Ca	5.42	6.34	1.27	15.57	13.48	12.22	4.61	7.99	8.49	0.42	7.35
Mn	0.15	0.13	0.29	-	0.22	0.18	-	0.22	-	-	0.17
Fe	2.01	3.99	42.19	7.56	2.16	24.15	3.84	45.67	43.91	32.46	4.45
Zn	2.25	3.81	3.99	2.81	7.38	3.42	3.71	1.93	1.86	2.99	5.81
Pb	7.47	6.24	-	4.75	5.54	1.82	1.62	-	-	-	3.45
Mg	-	0.54	-	0.85	1.09	1.40	0.68	4.95	4.58	8.79	-
Ti	-	0.36	-	0.22	0.33	0.45	3.32	-	-	-	3.84
Cu	-	-	-	0.36	-	1.23	0.17	-	-	0.26	0.26
Ni	-	-	-	-	-	0.19	-	-	-	-	-
Cr							0.37				0.36

## Data Availability

The data presented in this study are available upon request from the first author.

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
