# Peer review of "Effect of Flue Gases’ Corrosive Components on the Degradation Process of Evaporator Tubes"

_materials, 2021, doi:10.3390/ma14143860_

Round 1

Reviewer 1 Report

Gas corrosion of steel is a well-studied phenomenon. There are known thermodynamic and kinetic physical relationships of the formation of corrosive products at various temperatures and compositions of the  flue gases. The article is a report on the cause of boiler tubes failure due to adhesive wear under the agressive influence of flue gases. Obviously, this work was carried out as a commercial order from the enterprise. The authors used minimal research capabilities (metallography and local X-ray spectral analysis). However, this made it possible to determine the chemical composition of flue gases and scientifically substantiate the failure of boiler tubes made of steel 1.1148. The scientific value of this article is minimal. Having a sufficient amount of data on reducing the pipe thickness under specific operating conditions, the authors did not make a temperature-time analysis of corrosion. Having metallographic data, the authors could not even determine the depth of the decarburized layer on the outer surface of the pipe.

All the obtained results are scientifically substantiated, and they made it possible to determine the chemical composition of flue gases by corrosion products. Conclusions are useful for municipal needs.

Please indicate the type of scanning electron microscope.

Author Response

   Dear Reviewer,

Thank You for the review of our submitted manuscript. Individual reviewers had various comments on some parts of the article (even contradictory). We have tried to make changes and improvements in the article based on the compromise and your comments as follows:

Comments and Suggestions for Authors

Gas corrosion of steel is a well-studied phenomenon. There are known thermodynamic and kinetic physical relationships of the formation of corrosive products at various temperatures and compositions of the flue gases. The article is a report on the cause of boiler tubes failure due to adhesive wear under the aggressive influence of flue gases. Obviously, this work was carried out as a commercial order from the enterprise. The authors used minimal research capabilities (metallography and local X-ray spectral analysis). However, this made it possible to determine the chemical composition of flue gases and scientifically substantiate the failure of boiler tubes made of steel 1.1148. The scientific value of this article is minimal. Having a sufficient amount of data on reducing the pipe thickness under specific operating conditions, the authors did not make a temperature-time analysis of corrosion. Having metallographic data, the authors could not even determine the depth of the decarburized layer on the outer surface of the pipe.

All the obtained results are scientifically substantiated, and they made it possible to determine the chemical composition of flue gases by corrosion products. Conclusions are useful for municipal needs.

Please indicate the type of scanning electron microscope.

Response - Review Report:

The type of scanning electron microscope is JEOL JSM 7000F. We have added this to chapter 2. Materials and Methods, line 193.

Thank you for your time and for the consideration as well as the finest suggestions in order to improve scientific quality of our article.

   Best regards,

     Authors

Reviewer 2 Report

This paper focuses on the analysis of the influence of flue gas on the outer surface of evaporator tube and the influence of saturated steam on the inner surface of evaporator tube. The work is detailed and the theoretical support is sufficient. The reason of evaporator structure damage is analyzed. It is suggested that we accept it with a little modification. Here are some suggestions for the author. 

  1. There is a lack of comparison with similar studies in the past.
  2.  The picture background is messy and some pictures are not bright enough. It is suggested to modify the pictures. 
  3. Reduce some unnecessary repetition, increase the analysis and induction of test result data.
  4. In the second part, Materials and Methods, the test of tensile test can only explain the result of steel tube perforation, but it is not the cause.
  5. In Table 3, only 1245 of the damage properties were marked, but there was no 3? 
  6. The SEM image of Figure 18 is not dimensioned .

Author Response

   Dear Reviewer,

Thank You for the review of our submitted manuscript. Individual reviewers had various comments on some parts of the article (even contradictory). We have tried to make changes and improvements in the article based on the compromise and your comments as follows:

Comments and Suggestions for Authors

This paper focuses on the analysis of the influence of flue gas on the outer surface of evaporator tube and the influence of saturated steam on the inner surface of evaporator tube. The work is detailed and the theoretical support is sufficient. The reason of evaporator structure damage is analyzed. It is suggested that we accept it with a little modification. Here are some suggestions for the author.

1.

There is a lack of comparison with similar studies in the past.

Response - Review Report:

We compared our obtained results with similar studies in the past and included them in Chapter no. 4 - Discussion.

2.

The picture background is messy and some pictures are not bright enough. It is suggested to modify the pictures.

Response - Review Report:

Accepted.

Figures 1., 3. and 17. were removed, Figure 2. (now Figure 1.), Figure 4. (now Figure 2.) and Figure 13. (now Figure 11.) are modified.

3.

Reduce some unnecessary repetition, increase the analysis and induction of test result data.

Response - Review Report:

Accepted.

The unnecessary repetition was reduced, and the analysis and the test result data processing were improved.

4.

In the second part, Materials and Methods, the test of tensile test can only explain the result of steel tube perforation, but it is not the cause.

Response - Review Report:

Accepted.

A tensile test was used for the determination of the mechanical properties of steel. We have included this information in the text, Chapter 2. Materials and Methods, line 176.

5.

In Table 3, only 1 2 4 5 of the damage properties were marked, but there was no 3?

Response - Review Report:

Despite the original intention to analyze, we did not use Cutout no. 3, but we were left with the original sample markings. However, we accept your suggestion, and therefore we renumbered the cutouts according to the logical sequence.

6.

The SEM image of Figure 18 is not dimensioned.

Response - Review Report:

The scale was accidentally cropped while inserting the image into the template. This was fixed, and the SEM image (now Figure 15.) is dimensioned.

Thank you for your time and for the consideration as well as the finest suggestions in order to improve scientific quality of our article.

   Best regards,

     Authors

Reviewer 3 Report

The manuscript needs to be corrected:

  • There is no statement of the research goal and the task to achieve it. It is necessary to expand the review, consider in more detail the studies conducted (reference with a large range [8-13] in line 60 and [15-20] in line 73) and formulate the problem of the study (for example, the wear of the elements of the evaporators made of steel W. Nr. 1.1148 with a 31-year service life, or (and) there are no comprehensive studies of the effect of flue gases (specify the type) and the degree of purity of the water coolant, etc.).);
  • Photos of Figure 1, 3, and 4 need to be redone or replaced. Now they are poorly perceived, i.e. in the first figure the surfaces before the bend of the pipe are marked, and in the third and fourth, as it were, the section of the pipe before its bend is examined;
  • Clarify the data in Tables 1 and 2. Now the first one shows Re/Rp0. 2, and the second one shows Rp0. 2 (how to compare?). Also, in the first table, for some reason, the minimum pressure is indicated only in the first column;
  • The novelty of the presented studies is largely related to the identification of new patterns of the complex influence of flue gases of a particular type and the degree of purity of the water coolant during the long service life of the steel evaporator, but this is not reflected in the title and keywords. This must be reflected in the annotation and conclusion, indicating the limits of the content of Na, R, Cl in gases and Mg, Ca, Na and Si in water;
  • In Table 3, for some reason, there is no designation " 3 "for the variant" inner surface of tube (under the deposit)", although it is considered in the manuscript (see Figure 9);
  • It is mandatory to indicate the novelty instead of the first four well-known conclusions, i.e., what was first obtained as a result of this work, and the fifth should contain the limits of the concentrations of chemical elements and quantitative indicators of their influence on the wear of the evaporator elements.

Author Response

   Dear Reviewer,

Thank You for the review of our submitted manuscript. Individual reviewers had various comments on some parts of the article (even contradictory). We have tried to make changes and improvements in the article based on the compromise and your comments as follows:

Comments and Suggestions for Authors

The manuscript needs to be corrected:

1.

There is no statement of the research goal and the task to achieve it. It is necessary to expand the review, consider in more detail the studies conducted (reference with a large range [8-13] in line 60 and [15-20] in line 73) and formulate the problem of the study (for example, the wear of the elements of the evaporators made of steel W. Nr. 1.1148 with a 31-year service life, or (and) there are no comprehensive studies of the effect of flue gases (specify the type) and the degree of purity of the water coolant, etc.)).

Response - Review Report:

Accepted.

The review of studies (references with a large range) was actualized in more detail (1. Introduction). Also, at the end of the Introduction, the research goal and the task of our study were formulated.

2.

Photos of Figure 1, 3, and 4 need to be redone or replaced. Now they are poorly perceived, i.e. in the first figure the surfaces before the bend of the pipe are marked, and in the third and fourth, as it were, the section of the pipe before its bend is examined.

Response - Review Report:

Accepted.

Figures 1. and 3. were removed, and Figure 4. (now Figure 2.) was modified.

Also, Figure 17. was removed, and Figure 13. (now Figure 11.) was replaced with the detailed one.

3.

Clarify the data in Tables 1 and 2. Now the first one shows Re/Rp0. 2, and the second one shows Rp0. 2 (how to compare?). Also, in the first table, for some reason, the minimum pressure is indicated only in the first column.

Response - Review Report:

Because steel has indistinctive yield strength, we have mentioned Rp0.2. The minimum value applies to all Rp0.2 values, and it is adjusted in Table 1 now.

4.

The novelty of the presented studies is largely related to the identification of new patterns of the complex influence of flue gases of a particular type and the degree of purity of the water coolant during the long service life of the steel evaporator, but this is not reflected in the title and keywords. This must be reflected in the annotation and conclusion, indicating the limits of the content of Na, R, Cl in gases and Mg, Ca, Na and Si in water.

Response - Review Report:

Accepted.

The title, abstract, conclusions, and keywords have been improved and include the new patterns of the complex influence of flue gases of a particular type and the degree of purity of the water coolant during the long service life of the steel evaporator already.

5.

In Table 3, for some reason, there is no designation " 3 "for the variant" inner surface of tube (under the deposit)", although it is considered in the manuscript (see Figure 9).

Response - Review Report:

Despite the original intention to analyze, we did not use Cutout no. 3, but we were left with the original sample markings. However, we accept your suggestion, and therefore we renumbered the cutouts according to the logical sequence.

6.

It is mandatory to indicate the novelty instead of the first four well-known conclusions, i.e., what was first obtained as a result of this work, and the fifth should contain the limits of the concentrations of chemical elements and quantitative indicators of their influence on the wear of the evaporator elements.

Response - Review Report:

Accepted.

The Conclusions have been improved and according to your suggestion.

Thank you for your time and for the consideration as well as the finest suggestions in order to improve scientific quality of our article.

   Best regards,

     Authors

Reviewer 4 Report

The authors report a study on the effects of flue gases and saturated steam on the surface of evaporator tube. The paper is well written and the reported results have relevance in the materials corrosion field.

I believe that the manuscript can be accepted for publication in this Journal after addressing some minor revisions.

In particular, the novelty of the work with respect to that reported by other authors in similar studies must be better discussed.

Moreover, the related advances in the field must be better addressed in the Conclusions section.

Author Response

   Dear Reviewer,

Thank You for the review of our submitted manuscript. Individual reviewers had various comments on some parts of the article (even contradictory). We have tried to make changes and improvements in the article based on the compromise and your comments as follows:

Comments and Suggestions for Authors

The authors report a study on the effects of flue gases and saturated steam on the surface of evaporator tube. The paper is well written and the reported results have relevance in the materials corrosion field.

I believe that the manuscript can be accepted for publication in this Journal after addressing some minor revisions.

1.

In particular, the novelty of the work with respect to that reported by other authors in similar studies must be better discussed.

Response - Review Report:

We compared our obtained results with similar studies in the past, included them in chapter no. 4 (Discussion) and have improved this part of our manuscript.

2.

Moreover, the related advances in the field must be better addressed in the Conclusions section.

Response - Review Report:

Accepted.

The Conclusions have been improved according to your suggestion.

Thank you for your time and for the consideration as well as the finest suggestions in order to improve scientific quality of our article.

   Best regards,

     Authors

Round 2

Reviewer 3 Report

In the output, be sure to indicate the quantitative indicators of the improved parameters!

Author Response

   Dear Reviewer,

We are grateful for your insight once more. We accepted your suggestion - the quantitative indicators were added to the Conclusions of our Manuscript.

Comments and Suggestions for Authors

In the output, be sure to indicate the quantitative indicators of the improved parameters!

Response - Review Report:

The quantitative indicators were added to the Conclusions.

   Best regards,

     Authors
